# Adipose derived stromal vascular fraction and fat graft for treating the hands of patients with systemic sclerosis. A randomized clinical trial

Martin Iglesias [1¤]*, Iván Torre-Villalvazo[2], Patricia Butrón-Gandarillas[1], Tatiana S. Rodríguez-Reyna[3], Erik A. Torre-Anaya[2], Martha Guevara-Cruz[2], Miguel A. Flores-Cháirez[4], Diana B. López-Contreras[4], Joana Y. López-Sánchez[4], Ángel J. Ruiz-Betanzos[5], Ana L. Méndez López[4], Carolina Rubio-Gutierrez[4], Fernando Téllez-Pallares[5], Fabian Nario-Chaidez[6]

1 Plastic Surgery Service at Instituto Nacional de Ciencias Medicas y Nutricion Salvador Zubiran, Mexico City, Mexico, 2 Nutrition Physiology Department at Instituto Nacional de Ciencias Medicas y Nutricion Salvador Zubiran, Mexico City, Mexico, 3 Rheumatology Department at Instituto Nacional de Ciencias Medicas y Nutricion Salvador Zubiran, Mexico City, Mexico, 4 Fellow-clerk in plastic surgery, Universidad Autonoma de Coahuila, Saltillo, Coahuila, Mexico, 5 Fellow-clerk in plastic surgery, Instituto Nacional de Ciencias Médicas y Nutricion Salvador Zubiran, Mexico City, Mexico, 6 Mesenchymal Stem cell Therapy Department at CBCells Biotechnology, Zapopan, Mexico

☯ These authors contributed equally to this work.
¤ Current address: Monte de Antisana 47, Jardines en la Montaña, Tlalpan, Mexico City, Mexico
* iglesias@drmartiniglesias.com

## Abstract

### Background

Systemic Sclerosis in the hand is characteristically evidenced by Raynaud's phenomenon, fibrosis of the skin, tendons, ligaments, and joints as well as digital ulcers with prolonged healing. Current medical treatment does not always cure these complications. Local adipose-derived stromal vascular fraction administration into the hands has been proposed as an emerging treatment due to its regenerative properties. The objective of this randomized controlled clinical trial was to evaluate the safety and clinical effects of fat micrografts plus adipose derived-stromal vascular fraction administration into the hands of patients with systemic sclerosis.

### Methods

This was an open-label, monocentric, randomized controlled study. Twenty patients diagnosed with systemic sclerosis were assigned to the experimental or control group. Fat micrografts plus the adipose derived-stromal vascular fraction were injected into the right hand of experimental group patients. The control group continued to receive only medical treatment. Demographic, serologic data and disease severity were recorded. Digital oximetry, pain, Raynaud phenomenon, digital ulcers number, mobility, thumb opposition, vascular density of the nail bed, skin affection of the hand, serologic antibodies, hand function, and quality of life scores were evaluated in both groups.

**Data Availability Statement:** All relevant data are within the paper and its Supporting Information files.

**Funding:** This research was supported by the Consejo Nacional de Ciencia y Tecnologia (CONACYT, https://conacyt.mx/) with the Sectorial Research Fund in Health and Social Security 10000/739/2017 (to MI). The funders had no role in study design, data collection and analysis, decision to publish, or preparation of the manuscript.

**Competing interests:** The authors have declared that no competing interests exist.

## Results

The results of the intervention were analyzed with the Wilcoxon rank test, and the differences between the control and experimental groups at 0 days and 168 days were analyzed with the Mann–Whitney U test. Adverse events were not observed in both groups. At the end of the study, statistically significant improvements were observed in pain levels ($p < 0.05$) and number of digital ulcers ($p < 0.01$) in the experimental vs control group.

## Conclusion

The injection of adipose derived-stromal vascular fraction plus fat micrografts is a reproducible, and safe technique. Pain and digital ulcers in the hands of patients with systemic sclerosis can be treated with this technique plus conventional medical treatment.

## Introduction

Systemic sclerosis (SSc) is characterized by the presence of functional and structural microvasculopathy resulting in ischemia that leads to cutaneous and visceral fibrosis. In the hand, this disease is characteristically evidenced by Raynaud's phenomenon (RP) and fibrosis of the skin, tendons, ligaments, and joints [1–4]. Further, the hand is affected in 47 to 97% of the SSc cases [5, 6]. Permanent ischemia leads to pain and the appearance of digital ulcers that result in prolonged healing durations, recurrences, and infections and may culminate in digital amputation. This musculoskeletal involvement is one of the main etiologies of devastating disability and a dramatic decrease in the quality of life of patients with SSc [6, 7]. The current treatment is the control of the disease with medications and rehabilitation [8]. However, these options control the symptoms temporarily but do not decrease the period of time required for ulcer healing or improve hand function. Recently, the use of adipose derived stromal vascular fraction (ADSVF) as a regenerative medicine approach has been proposed for control and/or prevention of such hand complications [9, 10].

Zuk et al., described the presence of adipose stem cells (ASCs) in the ADSVF, in 2002 [11], and this partially explains the regenerative properties of ADSVF. Additionally, it contains precursor and mature hematopoietic and endothelial cells, together with immune cells, and all these cells contribute to its regenerative properties [11–23]. The ADSVF administration has positioned itself as an alternative treatment for many pathologies and for immunological diseases that cause ischemia and cutaneous fibrosis, which are also the main manifestations of SSc [11–23].

Granel et al. administered the ADSVF into the hands of 12 patients who predominantly had limited cutaneous SSc with no severe involvement of the internal organs. Subsequently, they reported improvements in pain, frequency and intensity of RP and the healing of digital ulcers that were resistant to conventional treatment. There was also an improvement in strength, movement, Cochin Functional Scale Scores, and a decrease in the digital circumference [9]. Furthermore, these results were reported to persist for 12, 22, and 30 months postoperatively with additional improvements in nail appearance and health based on improved Scleroderma Health Assessment Questionnaire (SHAQ) score, the ability to participate in household activities, and the approval of the procedure by the patients themselves [23–25]. Subsequently, this same working group conducted a controlled clinical trial. Ringer Lactate was applied to the control group and ADSVF to the experimental group. They reported

significant improvement in Cochin Functional Scale Score in both groups, but without statistically difference between both groups at three and six months [26]. Similarly, Park et al., administered ADSVF in 18 patients with SSc. Only skin fibrosis, hand edema and quality of life were significantly improved, and some digital ulcers were healed 24 weeks after the ADSVF injection [10].

Bank et al. infiltrated decanted fat in order to treat RP into the base of the fingers and hands of patients with SSc and obtained extraordinary results in terms of improving pain and RP [27]. Del Papa et al. centrifuged fat and injected the intermediate layer in patients with SSc and active digital ulcers that were resistant to conventional treatment [16, 17]. Subsequently, Del Papa et al. reported on the same technique in a controlled study, in 2019, that the digital ulcers healed after an average time of 8 weeks with the reduction of pain and formation of new capillaries [28]

Lipoatrophy may develop in patients with SSc due to the loss of adipocytes and defective stem cell function [15, 29]. This is an additional reason for the production of digital ulcers on the back of the joints. However, there was no statistical difference in the number of cells that adhered, alteration in the phenotype or surface antigen expression between adipose derived stem cells of patients with SSc and adipose derived stem cells of healthy patients [15]. Thus, there is a good possibility that the hands of patients with SSc could be treated with ADSVF plus fat grafts harvested from proximal sites, such as the abdominal wall. In the abdominal wall the mesenchymal stem cells are five-fold more abundant than in other anatomic sites [30]. However, to our knowledge, there are no studies that used ADSVF plus fat grafts for the treatment of hand manifestations in SSc. For these reasons and because there has been only one controlled clinical trial using ADSVF and only one controlled clinical trial using centrifuged fat [28], we designed this controlled clinical trial with the hypothesis that the administration of ADSVF plus fat micrografts and conventional medical treatment will be a safe and reproducible technique to improve clinical manifestations and deformities in the hands of patients with SSc better than only medical treatment.

## Material and methods

### Trial design

This open-label, monocentric, not blinded, randomized controlled pilot study was approved by the Institutional Review Committee of Instituto Nacional de Ciencias Medicas y Nutricion Salvador Zubiran, at Mexico City with the reference SCI-1505-15/15-1, and was registered and available in ClinicalTrials.gov under the identifier, NCT04387825.

### Participants, eligibility criteria, and settings

The database of 120 patients diagnosed with SSc was accessed by the Rheumatology department, who were contacted by telephone to invite them to participate in this study. Thirty patients were accepted and were initially evaluated. Then, the patients who met the selection criteria were evaluated by Cardiology, Reproductive Physiology, Internal Medicine, Oncology, Rheumatology and Plastic Surgery to detect contraindications. Of them 20 patients were selected (Fig 1).

The epidemiologist and the principal author used a random number table to randomize patients. These patients were allocated into control or experimental groups by the remaining researchers. The allocation was 1:1 ratio. Informed consent according to declaration of Helsinki to participate in this study was obtained from all the participants (S1 and S2 Files). Both the experimental and control groups contained 10 patients each who were diagnosed with SSc according to the criteria of the American College of Rheumatology [31], and LeRoy-Medsger

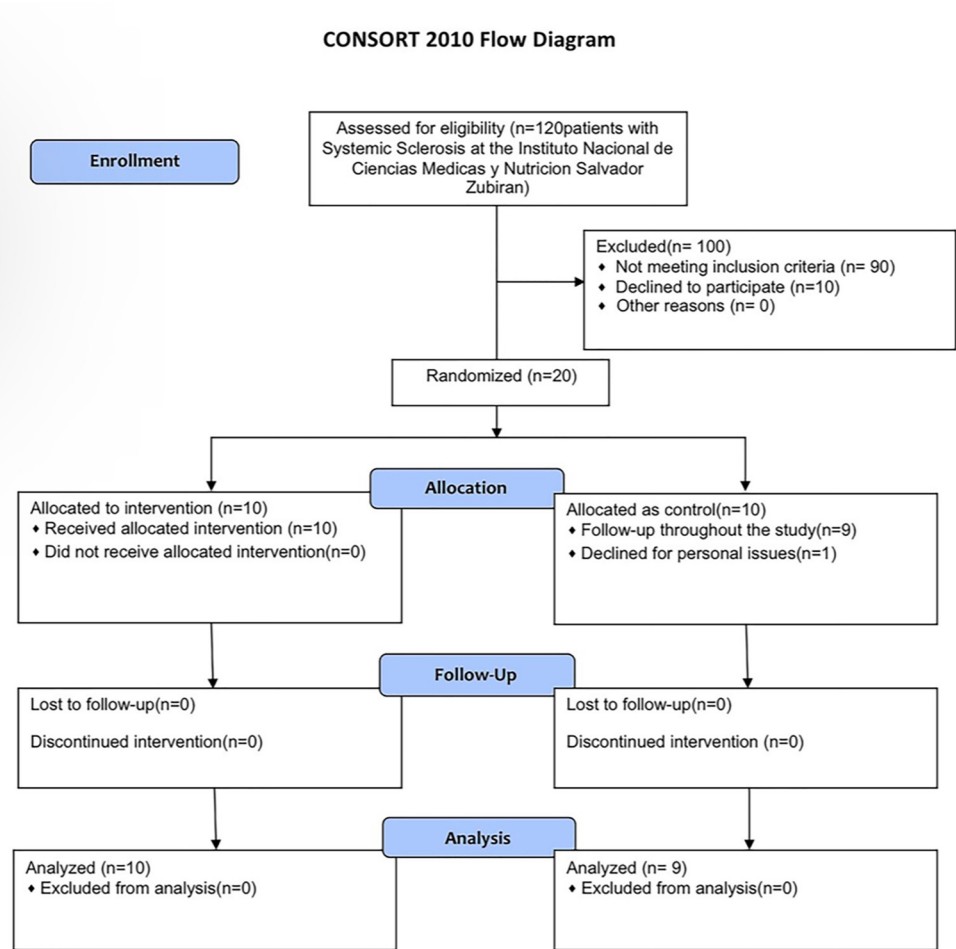

**Fig 1. Graphic of patients searching and selection.**

criteria [32]. All the patients were aged > 18 years and had BMIs > 18 kg/m². All the patients had undergone stable vasoactive and immunosuppressive therapies for at least 1 month before enrollment in the study, and these treatments were continued unchanged throughout the study period. The exclusion criteria were infected digital ulcers, comorbidities that could affect hand function, alcoholism, and/or drug abuse. The patients were evaluated in rooms with a controlled temperature of 24°C, and the study was performed from September 2018 to May 2019.

## Interventions

In the control group, the rheumatologic medical treatment prescribed and rehabilitation continued without any changes and the right hand was evaluated.

In the experimental group, it was decided that the ADSVF mix with micrografts would be administered to the most affected hand of the patients, which was the right hand in the entire group and the rehabilitation treatment was suspended until the end of the study. The ADSVF was processed by enzymatic digestion in a cellular therapy unit.

Rheumatological data were retrieved by the rheumatologist group, and medical interns were trained to allocate patients and collect the other data. Demographic and serologic data

**Table 1. Data evaluation.**

| Variable | Evaluation method | Follow up in days | | | | | | |
|---|---|---|---|---|---|---|---|---|
| | | Preop | 28 | 56 | 84 | 112 | 140 | 168 |
| Digital Oximetry | Transcutaneous Oximetry (%) | ✓ | ✓ | ✓ | ✓ | ✓ | ✓ | ✓ |
| Pain | Numeric scale from 1 to 10 | ✓ | ✓ | ✓ | ✓ | ✓ | ✓ | ✓ |
| Raynaud Phenomenon | Frequency of number of events per day/week; Duration of minutes in every event; intensity of the event (white, purple and red) | ✓ | ✓ | ✓ | ✓ | ✓ | ✓ | ✓ |
| Digital Ulcers | Number of ulcers | ✓ | ✓ | ✓ | ✓ | ✓ | ✓ | ✓ |
| Digital TAM* | Digital Manual Goniometry | ✓ | | | | | | ✓ |
| Thumb opposition | Kapandji Test | ✓ | | | | | | ✓ |
| Health status and disability index | SHAQ scale | ✓ | | | | | | ✓ |
| Hand Function | COCHIN scale | ✓ | | | | | | ✓ |
| Quality of Life | SF-36 scale | ✓ | | | | | | ✓ |
| Nail Capillaroscopic Patterns | Nailfold Videocapillaroscopy (early, active and late patterns) | ✓ | | | | | | ✓ |
| Skin affection of the hand | Modified Rodnan skin score (mRSS) | ✓ | | | | | | ✓ |
| Antibodies | ANA**, Anti-Topoisomerase I, Anti-centromere, Anti-U1-RNP | ✓ | | | | | | ✓ |

*TAM = Total Active Motion

**ANA = Anti-nuclear antibodies

were recorded as well as disease severity according to the Medsger Severity Scale. The evaluated variables and follow-up times are presented in Table 1.

Data regarding the presence of complications, such as compartment syndrome and/or infection, in both the donor site and treated hand were recorded daily for one week.

## Tissue collection and ADSVF preparation

Fat grafts were obtained by liposuction under local anesthesia, in a minor surgical procedures room equipped with necessary items for emergency situations. Briefly, the periumbilical region (9 patients) or posterior thigh (1 patient) was infiltrated with 500 ml of Klein's formula. Liposuction was performed with a 20-ml syringe and 3-mm diameter blunt cannula with 2-mm openings. A total of 100 ml of fat was harvested, from which 60 ml was transferred to sterile flask containing Hank's balanced saline solution with 5% albumin and was immediately transported to the laboratory. The adipose tissue was decanted for 10 minutes in a laminar flow cabinet for fraction separation. The upper fraction comprised disrupted adipocytes (oil), and the lower fraction was a fluid portion containing the anesthetic solution, erythrocytes, and dead cells. These both fractions were removed. The middle fraction was washed twice in phosphate-buffered saline (PBS) (Gibco, Thermo Fisher Scientific, Waltham, MA, USA) and subsequently incubated for 20 minutes in a type II collagenase solution (Sigma, St. Louis, MO, USA) in PBS at a concentration of 2 mg/ml of lipoaspirate. Allow the tubes to incubate at 37°C for 30 minutes with constant agitation. This was followed by addition of ethylenediaminetetraacetic acid (Sigma) in PBS. To separate the ADSVF from the floating adipocytes, this digested tissue was centrifuged at $800 \times g$ for 10 minutes. The compacted cells were resuspended in 3 ml of PBS and filtered through a 100-μm mesh cell strainer (Corning Inc., Corning, NY, USA). A 10 μl aliquot was obtained from each patient to characterize the ADSVF cellular identity, quantification, viability, and percentage of MSC by multi labeled flow cytometric assays. After the fat was processed, 2 ml of the ADSVF was transferred back to the minor surgical procedures room and was mixed with the 40 cc of fat in a 50 ml syringe. This mixture was homogenized

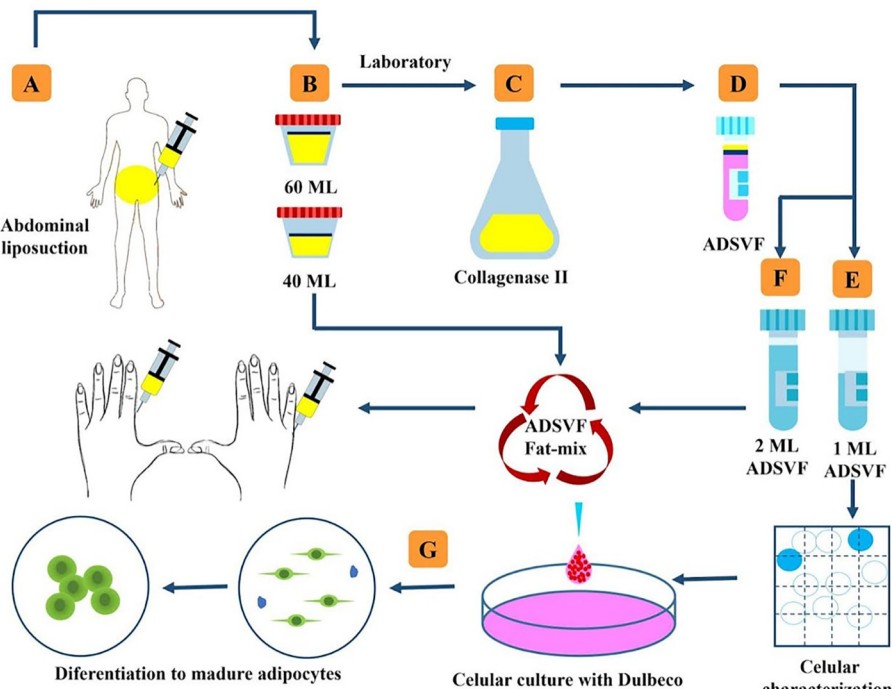

**Fig 2. Flow chart of the methodology.** (A) Fat grafts are obtained by abdominal liposuction. (B) A 60 ml volume was sent to the laboratory. (C) The enzymatic digestion of fat was performed by a collagenase separation method. (D) The initial adipose-derived stromal vascular fraction (ADSVF) was divided into two samples. (E) The first was a 1 ml sample for the characterization and control of the cell culture. (F) The second was a 2 ml sample that is reserved for administration to the patient in a lipoaspirate mixture by infiltration into the hand. (G) The differentiation process.

with a slight and a constant manual agitation of the syringe, later it was transferred to a 3 ml syringe for the autologous transplantation procedure. Another 1.0 ml sample was taken for cell culturing to determine the adipocyte differentiation capacity (Fig 2).

To verify that the ADSVF contained stromal cells, they were cultured and stimulated for their differentiation into mature adipocytes. The cells were seeded in 6-well culture plates in Dulbecco's modified Eagle's medium and were supplemented with fetal bovine serum and antibiotic/antifungal agents. Upon reaching 90% confluence, the cells were differentiated into mature adipocytes using a differentiation cocktail (2.5 μM dexamethasone, 0.5 mM 3-isobutyl-1-methylxanthine, 10 μg/ml insulin). As a control, a plate containing cells in the culture medium was left without adipogenic stimulation. After 20 days of differentiation, the cells were stained with red oil to assess fat accumulation (mature adipocyte phenotype), and the fat content in each culture was quantified by spectrophotometry (Fig 3).

## ADSVF administration into the hand

A nerve blocker was induced in the median, ulnar, and radial nerves at the wrist level using 2% lidocaine without epinephrine. Subsequently, 40 ml of fat was mixed with 2 ml of the ADSVF and was transferred to 1 ml and 3 ml syringes. Using a 11- sized scalpel blade, one incision was made at radial and ulnar edges of each of the metacarpophalangeal and interphalangeal joints [33]. With a 19-gauge blunt cannula (0.9 mm), 1.5 ml was administered along each radial and ulnar neurovascular digital pedicle through small radial and ulnar skin incisions at the level of the interphalangeal and metacarpophalangeal joints using the retro-tracing technique [33].

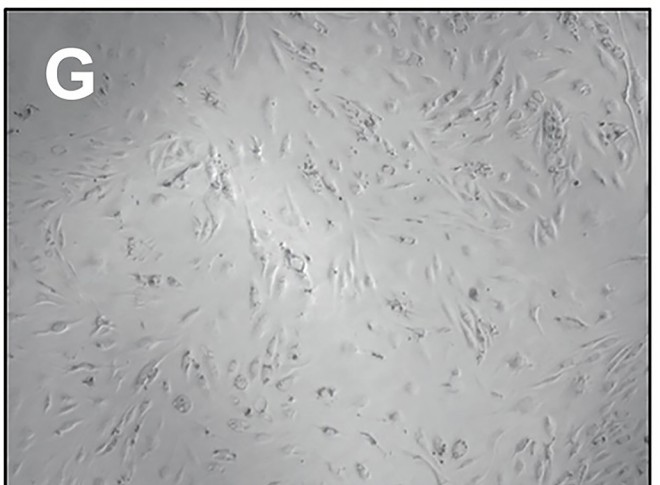
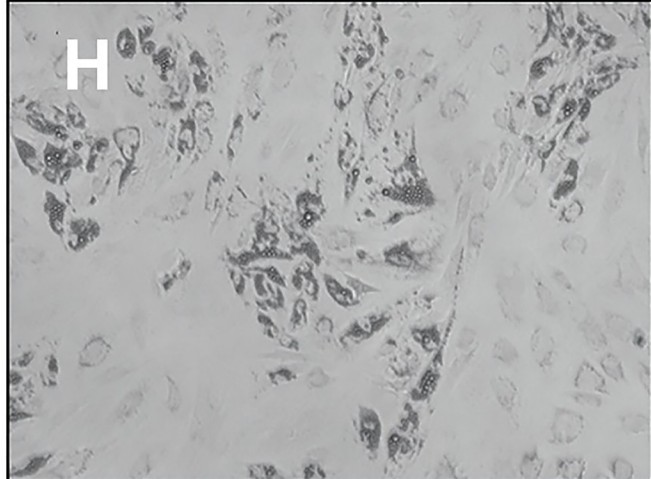
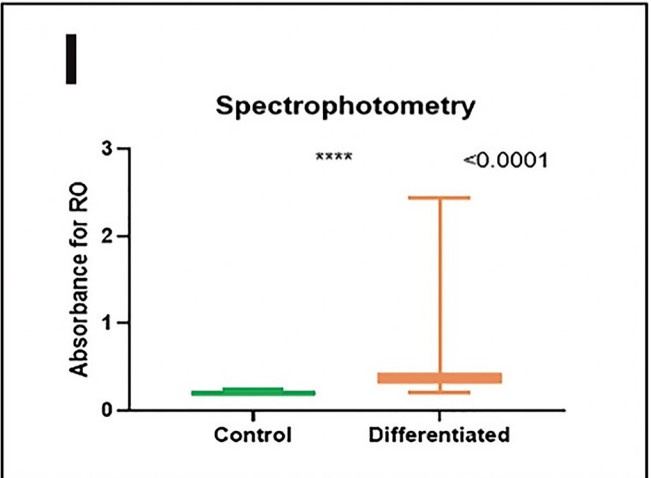

**Fig 3. Differentiation.** (G) Control culture without adipogenic stimulation stained with red oil (OR) at day 20. (H) Culture with adipogenic stimulation stained with OR on day 20. (I) Spectrophotometric analysis of the two groups depicted in a graphic. There was a higher OR absorption level in the differentiated group (p < 0.0001).

Additionally, 3 ml was administered to each side of the metacarpal trapezium joint, together with the subcutaneous distribution of 10 ml throughout the palm of the hand and even distribution of 10 ml in the back of the hand. This clinical trial did not have any deviation in the methodology described.

## Statistical analysis

Continuous variables are expressed as median with 95% CI with non normal distribution or mean with 95% CI with the normal distribution. Dichotomous variables are expressed as frequency and percentage. Categorical or dichotomous variables were analyzed with Fisher's exact test. The differences between before and after the intervention were analyzed with the Wilcoxon rank test, and the differences between the control and experimental groups at 0 days and 168 days were analyzed with the Mann–Whitney U test with non normal distribution and the student's t test of independent samples with normal distribution. Logarithmic transformation was performed before such analysis; subsequently, tests of normality (Kolmogorov-

Smirnof) and homoscedasticity (Levene's test) were applied to determine the nature of the data. The analysis was enhanced to determine the interaction between time (0/168 days) and treatment (control/experimental) with analysis of variance for repeated measures, and a p value of <0.05 was considered statistically significant of a tail. The data were analyzed using SPSS for Windows, version 24.00 (IBM Corp., Armonk, NY, USA) and GraphPad Prism software version 7 (GraphPad Software, San Diego, CA, USA).

## Results

Demographic, serological (antibodies), capillaroscopic pattern and disease severity-related data were similar between the two patient groups. All the patients presented with moderately intense vascular and joint conditions. Some patients exhibited severe internal organ involvement. One control group patient withdrew from the study due to personal problems (Table 2, S1 and S2 Tables).

The ADSVF acquisition and processing was constant. The total viable nucleated cell numbers and cellular characterization of the ADSVF are outlined in Table 3.

The abilities to adhere to plastic and differentiate to adipocytes are presented in Fig 3, greater amount of fat content was found in cell cultures induced toward adipocyte differentiation than in control cultures (p < 0.0001).

At the completion of ADSVF administration, the digits did not exhibit capillary filling, but circulation was reestablished itself after two hours without any treatment. There were no immediate or late adverse effects in the hands treated. In the donor site mild pain was reported by all patients and some of them had skin ecchymosis, which improved after 4–5 postoperative days. The total time of this procedure including harvesting of the fat grafts, ADSVF processing and its injection into the hands was 120 minutes.

Digital oximetry (SpO2), digital total active motion, thumb opposition, hand function, health status and disability index, and nail capillaroscopic pattern were similar in the two groups at the study's end.

In the experimental group, the pain score increased from 4.6 to 7.5 in the first week, gradually decreased to reach the basal value at day 21, and subsequently began to decrease further (p<0.01). In the control's group the pain decreased but without significance. For a better visual perception of pain evolution we have made a graphic representation (Fig 4). Significant improvements were observed in quality-of-life score (Short Form 36) in the experimental group at the end of the study (p<0.05). The frequency, intensity and duration of RP had significant improvement in both groups at 168 days but were more important in the experimental group (Table 4). The skin affection of the hand evaluated by mRSS, improved significantly in the control group (p<0.05) (Table 4).

However, when the results at 168 days were statistically compared between the groups, only pain exhibited a significant improvement (p < 0.05) (Table 4 and S3 Table).

The number of digital ulcers in the experimental group decreased within 28 days, while those in the control group increased within 84 days. In this way, the number of digital ulcers in the experimental group decreased significantly at the end of the study (p<0.01). During the follow-up period, ulcer recurrence was observed in one patient of the experimental group compared with three patients of the control group. (Table 5).

Clinically, the treated hands of the experimental group patients exhibited larger volumes, warmer temperatures, and faster capillary filling than the contralateral hands of the same patients (Fig 5 and S1 Video).

Both groups exhibited late capillaroscopic patterns. There was no significant change in the number of nailfold capillary loops between the groups at the study's end. One experimental

**Table 2. Demographic, serological and severity data.** Control and experimental groups.

| General Data | 0 days | | | 168 days | | |
|---|---|---|---|---|---|---|
| Patient (n) | Experimental | Control | P$^1$ | Experimental | Control | P |
| | 10 | 10 | | 10 | 9 | |
| Female n (%) | 10 (100%) | 9 (90%) | 0.99 | 10 (100%) | 8 (89%) | 0.47 |
| Age, mean (95%IC)* | 55.0(43.4,58.7) | 57.0(45.6,62.5) | 0.48 | 55.1(46.0,62.7) | 55.0(43.5,59.0) | 0.48 |
| Diffuse Sclerosis, n (%) | 8 (80%) | 6 (60%) | 0.62 | 8 (80%) | 6 (66.66%) | 0.62 |
| Antibodies | | | | | | |
| ANA positives, n (%) | 8 (80%) | 10 (100%) | 0.47 | 8 (80%) | 9 (100%) | 0.47 |
| Anti-Topoisomerase I, n (%) | 3 (30%) | 2 (20%) | 0.99 | 3 (30%) | 2 (22.2%) | 0.99 |
| Anti-centromere, n (%) | 3 (30%) | 3 (30%) | 0.99 | 3 (30%) | 3 (33.3%) | 0.99 |
| Anti-U1-RNP, n (%) | 2 (20%) | 3 (30%) | 0.99 | 2 (20%) | 3 (33.3%) | 0.62 |
| Organ Involvement | | | | | | |
| Vascular, n (%) | 10 (100%) | 10 (100%) | 0.99 | 8 (80%) | 9 (100%) | 0.47 |
| Severe vascular, n (%) | 6 (60%) | 7 (70%) | 0.99 | 2 (20%) | 4 (44.4%) | 0.35 |
| Articular, n (%) | 10 (100%) | 10 (100%) | 0.99 | 10 (100%) | 9 (100%) | 0.99 |
| Severe articular, n (%) | 6 (60%) | 2 (20%) | 0.17 | 3 (30%) | 3 (33.3%) | 0.99 |
| FTP, mean (95%IC) * | 3.15(2.30,3.71) | 4.00 (2.72,4.39) | 0.21 | 3.50(2.25,4.32) | 3.15(2.81,4.00) | 0.91 |
| Muscular, n (%) | 1 (10%) | 1 (10%) | 0.99 | 3 (30%) | 1 (11.1%) | 0.58 |
| Severe muscular, n (%) | 0 (-) | 0 (-) | 0.99 | 0 (-) | 0 (-) | 0.99 |
| Gastrointestinal, n (%) | 4 (40%) | 5 (50%) | 0.99 | 5 (50%) | 7 (77.7%) | 0.35 |
| Severe gastrointestinal, n (%) | 1 (10%) | 1 (10%) | 0.99 | 1 (10%) | 1 (11.1% | 0.99 |
| Pulmonary, n (%) | 6 (60%) | 9 (90%) | 0.30 | 6 (60%) | 9 (100%) | 0.08 |
| Severe pulmonary, n (%) | 1 (10%) | 1 (10%) | 0.99 | 1 (10%) | 1 (11.1%) | 0.99 |
| HAP, n (%) | 2 (20%) | 3 (30%) | 0.99 | 2 (20%) | 3 (33.3%) | 0.62 |
| Severe HAP, n (%) | 0 (-) | 0 (-) | 0.99 | 0 (-) | 0 (-) | 0.99 |
| Cardiac, n (%) | 1 (10%) | 1 (10%) | 0.99 | 1 (10%) | 1 (11.1%) | 0.99 |
| Severe cardiac, n (%) | 0 (-) | 0 (-) | 0.99 | 0 (-) | 0 (-) | 0.99 |
| Renal, n (%) | 0 (-) | 1 (10%) | 0.99 | 0 (-) | 1 (11.1%) | 0.47 |
| Severe renal n (%) | 0 (-) | 0 (-) | 0.99 | 0 (-) | 0 (-) | 0.99 |
| Capillaroscopic Pattern | | | | | | |
| Early phase n(%) | 2 (20%) | 1 (10%) | 0.99 | 3 (30%) | 0 (-) | 0.21 |
| Active phase n(%) | 0 (-) | 1 (10%) | 0.99 | 0 (-) | 1 (11.1%) | 0.47 |
| Late phase n(%) | 8 (80%) | 8 (80%) | 0.99 | 7 (70%) | 9 (100%) | 0.21 |

P$^1$ Statistical significance analyzed Fisher's exact test.

*Statistical significance analyzed with a student's t test of independent samples.

group patient exhibited an improved capillaroscopic pattern changing from late to early, while one control group patient exhibited a worsened capillaroscopic pattern changing from early to late (Table 2). The data collection can be consulted in the S4 Table.

## Discussion

The application of decanted fat [27], centrifuged fat [16], and the ADSVF [10, 17, 24, 34], has consistently and significantly improved pain, RP, and the healing of digital ulcers in the hands of patients with SSc, thereby improving their quality of life. Other inconsistently observed benefits, such as decreased digital circumferences, improvements in digital mobility and strength [35], improvements in the formation of new subungual capillaries, and improvements in function based on Cochin Functional Scale Scores, have also been reported [14, 15, 23]. Though,

**Table 3. Total viable nucleated cells and characterization of the ADSVF\*.**

| Total viable nucleated cell (x10$^6$) | 167.5 (39.8–543) |
| --- | --- |
| Cell viability (%) | 82.0 (59.3,95.1) |
| CD34+ (%) | 4.72 (0.50,18.2) |
| CD45+ (%) | 43.9 (1.39,67.5) |
| CD44+ (%) | 36.3(13.8,39.6) |
| CD73+ (%) | 6.18 (3.91,12.3) |
| CD90+ (%) | 34.4 (6.43,52.3) |
| CD105+ (%) | 7.27(1.19,54.9) |
| HLA-DR (%) | 12.1 (6.26,33.5) |
| Stromal cells (%) | 4.05 (2.51,6.83) |

\*ADSVF: Adipose Derived Stromal Vascular Fraction Median (min-max)

these previous studies have been uncontrolled with a short follow-up period of 6 months [10, 16, 17].

In controlled studies comparing the utility of ADSVF vs. placebo, the authors reported no statistically significant changes in Cochin Functional Scale Score, between the two groups [26, 28, 35]. However, they report certain trends of improvement in the Health Assessment Questionnaire Disability Index. The authors do not know the reason for this improvement in the placebo group, and some of them are contradictory results.

Prior to the authorization of this protocol, we treated two patients with SSc and severe pain with 1.5 cc micrografts of decanted fat applied along each digital neuro-vascular pedicle. They presented a significant decrease in pain. We cannot explain the reason for the decrease in pain, but it may be that the simple distension of the fibrous tissue around the nerves caused the pain to decrease. Perhaps these changes are similar to those observed by Daumas [26].

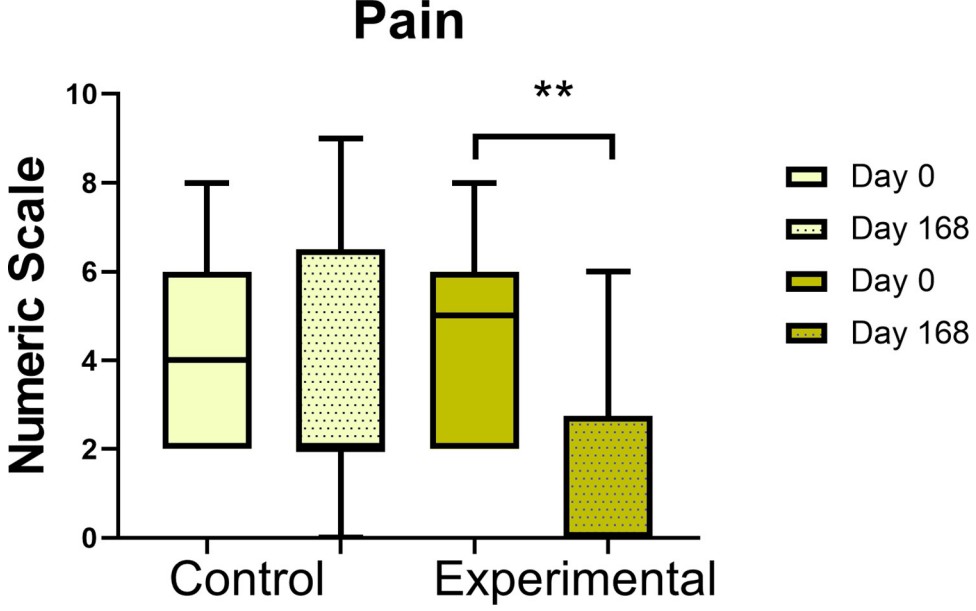

**Fig 4. Box plot of the pain scale before and after each group.** Statistical analysis was performed with Wilcoxon signed rank test, \*\*p < 0.01.

**Table 4. Results in control and experimental groups.**

| Concept | Group | Day 0 | Day 168 | P¹ | P² |
|---|---|---|---|---|---|
| Pain | Control | 4.00(2.64,6.16) | 2.00(1.51,6.04) | 0.91 | 0.02* |
| | Experimental | 5.00(3.08,6.12) | 0.00(0.00,2.95)* | 0.006* | |
| Quality of Life (SF-36) | Control | 40.0(28.5,46.4) | 35.0(17.5–51.3) 45.0(38.7–62.2) | 0.55 | 0.15 |
| | Experimental | 37.5(32.1–57.8) | | 0.04* | |
| Raynaud Phenomenon *Frequency n/week* | Control | 5.50(2.68,13.1) 4.50(2.87,6.33) | 00(0.00,03.18) 0.50(0.10,1.10) | 0.010* | 0.418 |
| | Experimental | | | 0.005* | |
| Raynaud Phenomenon *Intensity* | Control | 2.00(1.22,2.18) 2.00(1.50,2.10) | 0.00(0.00,1.44) 0.50(0.10,1.10) | 0.03* | 0.60 |
| | Experimental | | | 0.006* | |
| Raynaud Phenomenon *duration in minutes* | Control | 17.5(2.12,54.4) 12.5(6.55,35.6) | 0.00(0.00,10.3) 0.00(0.00,6.82) | 0.02* | 0.14 |
| | Experimental | | | 0.005* | |
| Skin affection of the hand (mRSS) | Control | 6.50(2.32,15.8)) | 5.50(1.42,12.9) | 0.05* | 0.19 |
| | Experimental | 15.0(6.84,17.7) | 15.5(6.36,16.6) | 0.07 | |

The sample size of the control group n = 9, experimental group n = 10; Data are presented as median (95% confidence intervals).

P¹ = Analyze the differences within each group between baseline and final results. Wilcoxon signed-rank test was used.

*p<0.05, U de Mann-Whitney with statistical significance.

P² = These data were log-transformed before statistical analysis was ANOVA for repeated measures to determine the time x group interaction. Analyze the differences between the groups.

Despite the different results in all these clinical studies, there is a statistical trend that the application of ADSVF may be useful for the treatment of hand complications caused by SSc.

The present controlled study with a 6-month follow-up period included patients with diffuse and limited SSc as well as moderate vascular and articular involvement, including some with severe internal organ involvement. The results observed in the experimental group were similar to those reported in uncontrolled studies. However, when the variables were compared with those of the control group, only the improvement in pain and number of digital ulcers were statistically significant. Only Raynaud's phenomenon and the quality of the skin on the hands improved significantly in the control group patients. We consider that these changes in the control group were due to medical treatment, rehabilitation, and the change in temperature to a warmer environment when the final evaluation of the study was carried out.

We apply fat micrografts with the aim of giving volume to the sclerotic fingers of patients with SSc. This volumetric effect of fat grafting has been well demonstrated [36]. These fat micrografts were applied to all fingers along each neurovascular bundle, to the palm and to the back of the hand, without causing permanent circulatory disturbances or compartment syndromes. Due to these probable consequences, most authors apply this cellular mixture only at the base of the fingers. In patients with Cochin Functional Scale Score above 30, the fingers exhibited structured joint stiffness and significant sclerodactyly. Even in these patients, it was possible to administer the cell suspension without causing adverse effects.

**Table 5. Digital ulcers in sclerosis systemic.**

| Group | Day 0 | Day 28 | Day 56 | Day 84 | Day 112 | Day 140 | Day 168 |
|---|---|---|---|---|---|---|---|
| Experimental (n 10) | 3 | 0 | 0 | 0 | 0 | 1 | 0 |
| Control (n 9) | 4 | 4 | 2 | 1 | 2 | 2 | 6 |
| p value* | 0.50 | 0.03** | 0.21 | 0.47 | 0.248 | 0.45 | 0.003** |

*statistical analysis was performed with the exact fisher test

** p<0.05

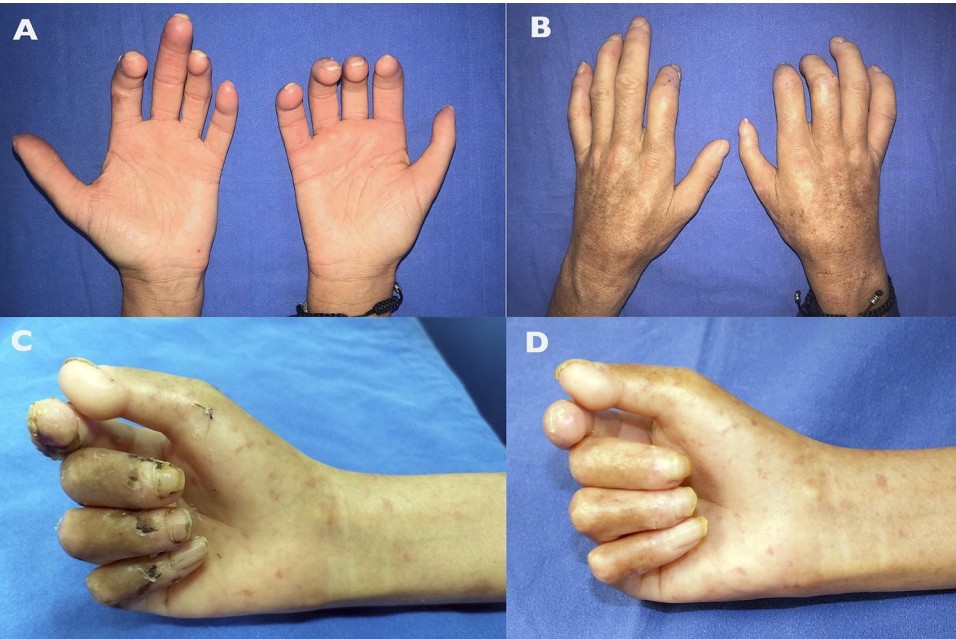

**Fig 5. Clinical results.** (A, B) Patient 1, female, age range 56–60 y/o. At the end of the study, the right hand was injected with micrografts enriched with the adipose-derived stromal vascular fraction. It exhibited a higher volume and was warmer than the left hand. (C) Patient 2, female with age range 31–35 y/o, clinical status before the treatment. The hand presented dermo-epidermal abrasions and digital ulcers. (D) The same patient with fewer digital ulcers, at the end of the study.

The mixture of fat micrografts plus ADSVF has previously been used in plastic surgery with the aim of increasing the integration of fat grafts. This procedure has been called cell-assisted lipotransfer [37]. For this reason we justify its use in these patients.

The treated hands of the experimental group patients exhibited a clinically greater volume, warmer temperatures, and faster capillary filling than the contralateral hands of the same patients and hands of the control group patients. Although these variables were not planned to be evaluated, they were clinical findings that we consider important to mention. This greater volume could prevent the recurrence of ulcers on the bony prominences.

The site and technique used for acquiring the lipoaspirate affect the quality of the ADSVF in terms of its use for regenerative treatment [38]. In our study, fat grafts were mainly obtained from the periumbilical region that is suggested to be the optimal site.

In this study, we isolated the ADSVF in a cell biology laboratory. The mean number of viable cells was three times higher compared to those processed by a closed system kit (Celution 800/CRS system and SmartXR kit o DongKoo Bio & Pharma Co., Ltd., South Korea) [9, 10, 35]. Even when there was a larger number of cells in the ADSVF, in addition to those in the fat micrografts, it did not affect our final results.

The techniques performed by the Granel and Magalon group for the characterization and determination of the proportion of the ASCs contained in the ADSVF were consistent with the requirements of the International Federation for Adipose Therapeutics and Science and the International Society for Cellular Therapy and with the Bourin report [9, 24, 35]. In the present study, cell characterization was performed similarly using six markers and human leukocyte antigen (HLA)-DR. Their values, together with their adherence to plastic and differentiation toward adipocytes in the laboratory, indicate that the cellular mixture in the ADSVF from our patients, which included mesenchymal stromal cells 4.05%, endothelial progenitor

cells, pericytes, hematopoietic, and immune cells, is similar to those reported previously [9, 22, 24, 38]. Therefore, the obtained results may be explained by the local angiogenic, anti-inflammatory, antifibrotic, immunomodulatory, an regenerative properties of the autologous injected ASCs. The scope of this study is not to describe the how and why of the regenerative effects of ADSVF.

It has been demonstrated that the vascular or regenerative properties of the ADSVF are not compromised by SSc [22]. Although it is assumed that the administration of the ADSVF may have some systemic effects, the involvement of internal organs did not improve or worsen in association with SSc in our patients.

We must point out that this study was carried out exclusively in Mexican mestizo patients and despite this, the results are similar to those previously reported. However, it is important to mention that in future studies ethnicity should be a variable to consider.

This study has the disadvantages of not having been a blind study for patients and researchers, as well as a limited number of patients studied. Although the number of digital ulcers was statistically lower in the experimental group compared to the control group at the end of the study, these results must be interpreted with caution, since unfortunately we did not assess the number of digital ulcers existing at the beginning and at the end of the study in each patient. The loss of a patient even though he belonged to the control group is a significant loss considering the size of the sample. Future studies should try to avoid these limitations.

## Conclusion

Our study has proven that when combined with conventional medical therapies, the administration of fat micrografts enriched with the ADSVF into the hands of patients with SSc can significantly improve pain without adverse effects. Studies with larger numbers of patients are necessary in order to confirm that this method can improve the number of digital ulcers, RP, digital perfusion, vascular density of the nail bed, and the function of the hand. Based on the aforementioned findings, the regenerative medicinal benefits of the ADSVF may be an essential aid for rheumatologists and plastic surgeons before or during the treatment of hand deformities in patients with SSc.

## Supporting information

**S1 Checklist.**
(PDF)

**S1 File. Protocol informed consent.**
(DOCX)

**S2 File. Patient involvement.**
(DOCX)

**S1 Table. Patient medications.**
(PDF)

**S2 Table. Patient comorbidities.**
(PDF)

**S3 Table. Complete results control and experimental groups.**
(PDF)

**S4 Table. Statistics.**
(XLSX)

**S1 Video. Hands.**
(ZIP)

## Acknowledgments

The authors thank Claudia Chavez-Muñoz, MD, for her scientific support.

## Author Contributions

**Conceptualization:** Martin Iglesias, Tatiana S. Rodríguez-Reyna.

**Data curation:** Martin Iglesias, Tatiana S. Rodríguez-Reyna, Miguel A. Flores-Cháirez, Ángel J. Ruiz-Betanzos.

**Formal analysis:** Martin Iglesias, Martha Guevara-Cruz.

**Funding acquisition:** Martin Iglesias, Fabian Nario-Chaidez.

**Investigation:** Martin Iglesias, Iván Torre-Villalvazo, Patricia Butrón-Gandarillas, Tatiana S. Rodríguez-Reyna, Erik A. Torre-Anaya, Diana B. López-Contreras.

**Methodology:** Martin Iglesias, Joana Y. López-Sánchez, Fabian Nario-Chaidez.

**Project administration:** Martin Iglesias.

**Supervision:** Martin Iglesias, Carolina Rubio-Gutierrez, Fernando Téllez-Pallares.

**Validation:** Martin Iglesias, Ana L. Méndez López.

**Writing – original draft:** Martin Iglesias.

**Writing – review & editing:** Martin Iglesias.

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
