## [Decision Letter · Decision Letter 0]

20 Mar 2023

PONE-D-23-00573Adipose derived stromal vascular fraction and fat graft versus medical treatment for treating the hands of patients with systemic sclerosis. A randomized clinical trialPLOS ONE

Dear Dr. Iglesias,

Thank you for submitting your manuscript to PLOS ONE. After careful consideration, we feel that it has merit but does not fully meet PLOS ONE’s publication criteria as it currently stands. Therefore, we invite you to submit a revised version of the manuscript that addresses the points raised during the review process.

The manuscript has been evaluated by four reviewers, and their comments are available below. They found your study interesting, but also raised a number of concerns around methodological aspects of the study, as well as the interpretation  and discussion of the results. Please revise the manuscript to carefully address all the concerns raised.

We look forward to receiving your revised manuscript.

Kind regards,

Dario Ummarino, PhD

Senior Editor

PLOS ONE

Journal Requirements:

2. We note that you have selected “Clinicl Trial” as your article type. PLOS ONE requires that all clinical trials are registered in an appropriate registry (the WHO list of approved registries is at      https://www.who.int/clinical-trials-registry-platform/network/primary-registries"" https://www.who.int/clinical-trials-registry-platform/network/primary-registries and more information on trial registration is at http://www.icmje.org/about-icmje/faqs/clinical-trials-registration/). Please state the name of the registry and the registration number (e.g. ISRCTN or ClinicalTrials.gov) in the submission data and on the title page of your manuscript.

a) Please provide the complete date range for participant recruitment and follow-up in the methods section of your manuscript.

b) If you have not yet registered your trial in an appropriate registry, we now require you to do so and will need confirmation of the trial registry number before we can pass your paper to the next stage of review. Please include in the Methods section of your paper your reasons for not registering this study before enrolment of participants started. Please confirm that all related trials are registered by stating: “The authors confirm that all ongoing and related trials for this drug/intervention are registered”. Please see http://journals.plos.org/plosone/s/submission-guidelines#loc-clinical-trials for our policies on clinical trials

"This research was supported by the Consejo Nacional de Ciencia y Tecnologia (CONACYT, https://conacyt.mx/) with the Sectorial Research Fund in Health and Social Security 10000/739/2017 (to MI)."

5. We note that Figures 2 and 5 in your submission contain copyrighted images. All PLOS content is published under the Creative Commons Attribution License (CC BY 4.0), which means that the manuscript, images, and Supporting Information files will be freely available online, and any third party is permitted to access, download, copy, distribute, and use these materials in any way, even commercially, with proper attribution. For more information, see our copyright guidelines: http://journals.plos.org/plosone/s/licenses-and-copyright.

a. You may seek permission from the original copyright holder of Figures 2 and 5 to publish the content specifically under the CC BY 4.0 license. 

6. We note that the original protocol that you have uploaded as a Supporting Information file contains an institutional logo. As this logo is likely copyrighted, we ask that you please remove it from this file and upload an updated version upon resubmission.

Reviewers' comments:

Reviewer's Responses to Questions

**Comments to the Author**

1. Is the manuscript technically sound, and do the data support the conclusions?

Reviewer #1: Yes

Reviewer #2: No

Reviewer #3: Partly

Reviewer #4: Partly

2. Has the statistical analysis been performed appropriately and rigorously? 

Reviewer #1: I Don't Know

Reviewer #2: No

Reviewer #3: Yes

Reviewer #4: Yes

3. Have the authors made all data underlying the findings in their manuscript fully available?

Reviewer #1: Yes

Reviewer #2: Yes

Reviewer #3: Yes

Reviewer #4: Yes

4. Is the manuscript presented in an intelligible fashion and written in standard English?

Reviewer #1: Yes

Reviewer #2: Yes

Reviewer #3: No

Reviewer #4: Yes

5. Review Comments to the Author

Reviewer #1: This is descriptive serie of cases comparing the injection of ADSVF mix with fat to treat hand disability of patients suffering from SSc to control group of patients that just follow their basic treatment.

The technique described in this paper combines two adipose derived treatments from fat applied to the hands and are inspired from Bank and Granel previously published studies.

The approach is interesting as it offers the possibility to treat the whole hand.

However, some points need to be clarified :

- the number of VNC / cc of fat is different in table 3 and line 400 in the discussion

- regarding this parameter, the yield is very high (> 2 millions / cc of fat considering that 60 cc is digested) compared to other published data (see Francois et al in 2020, Cells).. the author should discuss this point and precise the temperature of digestion and number of enzymatic unit used for the digestion.

- the author precise that 4% of the cells were ASC : how did they reach to this number ? By flow cytometry or using a CFU-F clonogenicity test

- the method used for flow cytometry seems to be monolabelled whereas a multi labelled as described in the paper of Granel could allow to determine all cell subtypes from ADSVF. This point should be discussed

- mixing 2 cc of ADSVF with 40 cc of fat seem difficult to guarantee the presence ADSVF cells within all the mixed product. The author could provide details on the technique used to perform the mix

- the discussion should include the paper from Khanna et al in 2022 in Arthritis Rheumatology

- the discussion should mention an important limit regarding methodology which is the absence of "double blind method". Indeed, the results from Daumas and Khannah in double blind RCT confirm the high placebo effect of this procedure.

Reviewer #2: This manuscript analyzes data generated from a controlled clinical trial evaluating the safety and clinical effects of "fat micrografts + adipose-derived stromal vascular function" group, versus the control group ("only medical treatment") in patients with systemic sclerosis. The study was registered as a RCT (with a legit NCT number), and was approved by the respective IRB/Ethics Committee. While the study objectives sound interesting, is important, and on target, some shortcomings were observed, in regards to abiding by the CONSORT guidelines for conducting and reporting results of high-quality randomized controlled trials (RCTs). Some other (statistical) comments were also provided.

1. Abstract: The Introduction needs a rewrite.

(a) In the sentence, "The objective of this controlled...", please mention BOTH the experimental group and the control group.

(b) Statistically significant results in the Abstract should be accompanied by appropriate p-values.

2. Methods:

Methods reporting need some work. An orderly manner is suggested, following CONSORT guidelines, without repeating information, such as Trial Design, Participant Eligibility Criteria and settings, Interventions, Outcomes, sample size/power considerations, Interim analysis and stopping rules, Randomization (details on random number generation, allocation concealment, implementation), Blinding issues, etc, should be mentioned. The authors are advised to create separate subsections for each of the possible topics (whichever necessary), and that way produce a very clear writeup. I see the Authors indeed made an attempt; however, they are advised to write it carefully, following nice examples in the manuscript below:

https://www.sciencedirect.com/science/article/pii/S0889540619300010

Specific comments:

(a) For instance, the randomization and allocation concealment should be made very clear (they are NOT the same thing); the trial staff recruiting patients should NOT have the randomization list. Randomization should be prepared by the trial statistician, and he/she would not participate in the recruiting.

(b) Sample size/power: It appeared strange to find no paragraph of the desired sample size/power for the study (given that this is analysis of data generated from a randomized trial). Formal power calculation should be presented, should focus on the primary response variable, at some desired effect size, and say at 5% level of significance.

(c) Statistical Analysis: Mention clearly, why nonparametric assessments (Wilcoxon range test, and Mann-Whitney U tests) were chosen wrt. analysis, bypassing parametric modeling, initially. Furthermore, in the longitudinal evaluation, repeated measures ANOVA was used, which is strictly based on Gaussian assumptions. How were the assumptions checked? If those fail, please resort to nonparametric analysis, such as the Friedman's test, etc.

I have an additional question. Given that the study is longitudinal, with covariates measured either at baseline, or at various time-points, why was a formal longitudinal analysis not conducted, via a linear mixed model, or GEE.

3. Results & Conclusions:

(a) The authors should check that any statement of significance should be followed by a p-value in the entire Results section.

(b) The Discussion section should clearly state that the findings of this study is only from a RCT of Mexican subjects. Hence, future studies (using subjects recruited at other locations/country) are needed to validate the current findings.

Reviewer #3: In this manuscript, the authors present data from a clinical studying investigating the efficacy of utilizing lipoaspirate mixed with stromal vascular fraction to assist pain related debilities due to systemic sclerosis. The study cohorts consisted of 10 patients receiving continuous standard of care treatment and 10 patients receiving lipo+SVF, all in the right hand, which was determined to be the most severely affected. Results indicated that lipo+SVF significantly reduced patient perception of pain.

Major critiques:

1. This is the first noted report of use of lipoaspirate with SVF supplementation to treat systemic sclerosis related hand debility, but the logical jump to utilizing this therapeutic strategy is unclear. Though reported use of SVF demonstrated acute effects, results were not durable, thus alternative strategies were needed. However, two discussed reports of use of fat injections for Ssc seemed to have sustained results, thus it is unclear which aspect of these protocols require improvement. As previous reports suggest long term, durable results were achieved when treated Ssc related hands with lipoaspirate (condensed or decanted), what is the rationale for modifying lipoaspirate in this study?

2. The primary hypothesis for admixing SVF with lipoaspirate in this clinical study is that adipose derived from Ssc patients incurs pathogenic related stem cell deficiencies in native tissue, thus supplementation is necessary. However, the ratio of ASC per gram of lipo in donors is not reported in this study and it is not clear if the naive lipo is deficient in stem cells. Further, the authors determined the total number of nucleated cells added to lipoaspirate prior to injection, yet, it seems no significant correlation existed between supplemented values and measured outcomes. Therefore, it is unclear what value SVF supplementation adds to use of lipoaspirate alone.

3. Development of a surgical or therapeutic approach to improve patient quality of life and reduce hand disability is a critical unmet need, however a future opportunity for a more robust study would be ASC supplementation in a randomly selected hand compared to fat alone in the contralateral, such that patients served as their own internal control to measure the benefit of SVF supplementation. The process of isolation SVF is time consuming and expensive, requiring patients to be exposed to increased duration of anesthesia and surgical risk. Thus, it is extremely important to adequately assess the necessity of the SVF isolation procedure and supplementation. However, as reported herein, this study does not make clear or not, the necessity of SVF supplementation to achieve durable results.

Minor criticisms:

1. The term Wilcoxon "Range" test is used multiple times. Do the authors mean Wilcoxon "Rank" test meant?

2. The authors measured clinical outcomes in the treated hand and had an untreated hand in the same patient which could have served as an internal baseline for the effects of continued medical treatment. In essence, what effects were seen in untreated contralateral hands?

3. More information needs to be provided as to how lipoaspirate was condensed or prepared in the OR prior to reinjection? When lipo washes were performed, which devices or methods were used? How was lipo condensed in the OR and exactly how was the SVF mixed with fat prior to injection?

Reviewer #4: The randomized clinical trial investigates the effect of the stromal vascular fraction with fat grafting for local manifestations in the hands of patients with systemic sclerosis. The reviewer applauds that the authors have performed a randomized clinical trial as there is a lack of randomized study designs in the field of plastic surgery and especially stem-cell enriched fat grafting.

Despite the study design, the manuscript should undergo major revisions before acceptance.

Title

1) As both groups continue their medication, I do not believe that ADSVF vs. medical treatment is an appropriate wording. I suggest removing “versus medical treatment” from the title.

Abstract

2) Headlines (Background, Methods, Results and Conclusion) would make the abstract more presentable.

Methods

3) Please report the reasons for declination of participation. As 100 patients declined it would be relevant to explore a potential selection bias.

4) Please clarify how the outcomes were collected. Which outcomes were self-reported? How often did the patients meet for clinical controls?

5) In line 117 it is stated that a broad group of physicians were involved to detect contraindications. What where the contraindications of the 10 excluded patients?

6) The period of the study is repeated in in line 146 and 202 where September and October are both mentioned as the starting month.

7) In general, I would suggest using months as time points as 168 days seems more arbitrary.

Statistics

8) In line 206 it is reported that continuous variables are reported as median with 95%CI. It should either be median with IQR or mean with 95%CI depending on the normal distribution.

9) Age and FTP are reported as means with 95%CI suggested a normal distribution, however the mentioned statistical tests are usually applied for non-normal continuous outcomes.

10) Where there any patients with multiple ulcers and if yes how was this handled statistically?

Results

11) The results section could be more structured as there are many different outcomes. Either start with the significant results or chose another more appropriate order.

12) When reporting p-values it should be clearer whether the analysis is a within group versus between group analysis.

13) In table 2 I would suggest not to perform statistical comparisons in groups without any values (e.g. severe cardiac involment) as a comparison of zero-values is not meaningful.

14) All p-values should be reported as less than instead of equals and follow the system of p<0.05, p<0.01, p<0.001 and p<0.0001. Non-significant p-values should be exact.

15) In the legend it states that the reported p-values in p1 are both within and between group analyses but only two p-values are reported (if both within and between group analyses are reported I would expect three p-values?)

16) There is a mismatch between the values provided in table 4 and figure 4. E.g. the pain score in the experimental group is reported as 5.0 and 0 in table 4 but is visually assessed to be 4.5 and 1.5 in figure 4.

Discussion

17) I would suggest shortening the discussion and be more selective and concise when commenting your own results with the literature as reference.

18) I suggest beginning the discussion with the key results from this study with a subsequent comparison with the literature.

19) There should be a limitation section. Despite the randomized design the study is still of low quality due to a small sample size, no blinding of patients, surgeons or outcome assessors, no sham-procedure, loss to follow-up of 10% in one group and no power-calculation. These limitations should be stated.

20) In line 372-374: Why did the medical treatment (which was also continued in the experimental group), the warmer environment not affect the experimental group?

6. PLOS authors have the option to publish the peer review history of their article (what does this mean?). If published, this will include your full peer review and any attached files.

Reviewer #1: No

Reviewer #2: No

Reviewer #3: No

Reviewer #4: No

---

## [Author Response · Author response to Decision Letter 0]

11 May 2023

Dear editor, we appreciate the work that you and the reviewers have invested in considering our article entitled “Adipose derived stromal vascular fraction and fat graft versus medical treatment for treating the hands of patients with systemic sclerosis. A randomized clinical trial” in your journal. We attach the requested responses to the comments. 

Senior editor comments (May 5th, 2023)

Dear editor, thank you for your time and your comments.

We have uploaded all the data generated by the development of the protocol as Supporting Information Files to the PlosOne journal and we have added two additional files, as: S2 File. Patient involvement.

All parts of Figure 2 were created and illustrated with ideas from authors Erik Torre, Martin Iglesias, and Carolina Rubio. The Figure 2 has not been previously copyrighted, so we do not require a Request for Permission to Publish Content under CC-BY License. Therefore, we grant full permission for Figure 2 to be freely used to download, copy, distribute and use in any way by the journal and by third parties.

Senior editor comments ( March 20th, 2023)

The manuscript is structured according to the requirements of the PLOS ONE journal. The date range for participant recruitment and follow-up are written in the material and methods section of our manuscript. As well as clinical trials the authors confirm that all ongoing and related trials for these interventions are registered in clinicaltrial.gov (NCT04387825). We have added to the founding statement; there was no additional external funding received for this study, and we have also added this statement to the cover letter along with all supporting information files, were uploaded to this journal. Figure 2 has been created by the authors so it does not require copyright. Written permission from the patients from Figure 5 was uploaded. The logo in the protocol document in the supporting information has been removed.

Reviewer #1

-The number of VNC / cc of fat is different in table 3 and line 400 in the discussion 

Response:

Dear reviewer, we apologize for our mistake, the correct number of VNC is the one that´s on table 3, 167.5 (39.8-543). We have corrected it in the manuscript. 

“The mean number of viable cells in the ADSVF that we isolated was 167.5 x 106 (39.8-543) compared with 50.5 ± 23.8 (16.7.92.6) [9], and 42.1 x 106 ± 5.06 [10], cells in previous studies”.

-Regarding this parameter, the yield is very high (> 2 millions / cc of fat considering that 60 cc is digested) compared to other published data (see Francois et al in 2020, Cells).. the author should discuss this point and precise the temperature of digestion and number of enzymatic unit used for the digestion. 

Response:

The ADSVF used in patients with SSc has been processed in closed mechanical systems (Celution 800/CRS system, SmartXR kit; DongKoo Bio & Pharma Co., Ltd., South Korea) and none by enzymatic digestion. Although the results are inconstant, most authors agree that the number of VNC/cc is higher with enzymatic digestion. The amount of type II collagenase used was 2 mg/ml of lipoaspirate. The cell laboratory temperature at which the ADSVF was processed was 22 °C and the incubation of the lipoaspirate tubes with collagenase was 37 °C.

We have added the following line in the material and methods section:

“Allow the tubes to incubate at 37 °C for 30 minutes with constant agitation”

-The author precise that 4% of the cells were ASC : how did they reach to this number ? By flow cytometry or using a CFU-F clonogenicity test 

Response:

The best method to determine the concentration of mesenchymal cells in the ADSVF is by CFU-F. However, cytometry gives us an approximate number, although it is not real. The method used in this article to determine the number of mesenchymal cells is by flow cytometry, which was performed immediately after obtaining the ADSVF. The objective of this work was not to determine the number of mesenchymal cells in the ADSVF.

-The method used for flow cytometry seems to be monolabelled whereas a multi labelled as described in the paper of Granel could allow to determine all cell subtypes from ADSVF. This point should be discussed

Response:

The flow cytometry used in our study was multillabeled, similar to reported by Granel, which allowed us to determine the cell characterization.

We have corrected the paragraph as follows:

“A 10 μl aliquot was obtained from each patient to characterize the ADSVF cellular identity, quantification, viability, and percentage of MSC by multi labeled flow cytometric assays.”

-Mixing 2 cc of ADSVF with 40 cc of fat seem difficult to guarantee the presence ADSVF cells within all the mixed product. The author could provide details on the technique used to perform the mix 

Response:

We have modified the follow paragraph:

“After the fat was processed, 2 ml of the ADSVF was transferred back to the minor surgical procedures room, and was mixed with the 40 cc of fat in a 50 ml syringe. This mixture was homogenized with a slight and a constant manual agitation of the syringe, later it was transferred to a 3 ml syringe for the autologous transplantation procedure. Another 1.0 ml sample was taken for cell culturing to determine the adipocyte differentiation capacity (Fig 2).”

-The discussion should include the paper from Khanna et al in 2022 in Arthritis Rheumatology

Response:

Dear reviewer, thank you for your recommendation, we have added a new paragraph in discussion section and also the reference in this way:

“Khanna et al from the Michigan University, in a double-blind randomized trial conducted on 88 patients with diffuse and limited SSc, injected ADSVF processed with the Celution System to the fingers of these patients. The authors reported no statistically significant changes in hand function according to CHFS between the two groups. However, they report certain trends of improvement in the Health Assessment Questionnaire Disability Index in patients with diffuse sclerosis for the group treated with ADSVF. Their findings add to the results of previous studies and ours, demonstrating that the application of ADSVF can be a useful treatment for hand complications in these patients [35].” 

- The discussion should mention an important limit regarding methodology which is the absence of "double blind method".Indeed, the results from Daumas and Khannah in double blind RCT confirm the high placebo effect of this procedure.

Response:

Our protocol was approved in August 2015, when there were only two published articles. Our patients presented a Cochin Scale that varied from 4 to 63, with an average of 27. Due to the uncertainty of the benevolence of the procedure, the research committee of the Instituto Nacional de Ciencias Medicas y Nutricion Salvador Zubiran considered it important not to take risks in the extraction of fat and in the application of ADSVF in the control group. Even though our study is not blind, it is a randomized, strict clinical study that shows similar results to those of double-blind clinical studies. 

As also requested by another reviewer, we have added the following paragraph:

“This study has the disadvantages of not having been a blind study for patients and researchers, as well as a limited number of patients studied. The loss of a patient even though he belonged to the control group is a significant loss considering the size of the sample. Future studies should try to avoid these limitations.” 

Reviewer #2

1. Abstract: The Introduction needs a rewrite.

(a) In the sentence, "The objective of this controlled...", please mention BOTH the experimental group and the control group.

Response:

Dear reviewer, thank you. We have added to the abstract the word “RANDOMIZED” due that we have a word limit in this section.

“The objective of this randomized controlled clinical trial was to evaluate the safety and clinical effects of fat micrografts plus adipose derived-stromal vascular fraction administration into the hands of patients with systemic sclerosis.”

(b) Statistically significant results in the Abstract should be accompanied by appropriate p-values.

Response:

We have added the p-values in the abstract:

“At the end of the study, statistically significant improvements were observed in pain levels (p=0.02) and number of digital ulcers (p=0.003) in the experimental vs control group.”

2. Methods:

Methods reporting need some work. An orderly manner is suggested, following CONSORT guidelines, without repeating information, such as Trial Design, Participant Eligibility Criteria and settings, Interventions, Outcomes, sample size/power considerations, Interim analysis and stopping rules, Randomization (details on random number generation, allocation concealment, implementation), Blinding issues, etc, should be mentioned. The authors are advised to create separate subsections for each of the possible topics (whichever necessary), and that way produce a very clear writeup. I see the Authors indeed made an attempt; however, they are advised to write it carefully, following nice examples in the manuscript below:

Response: 

Dear reviewer, we have followed your suggestion and the structure of the material and methods section has been modified. Due to the extensiveness of the corrections, I kindly ask you to review them in the manuscript.

Specific comments:

(a) For instance, the randomization and allocation concealment should be made very clear (they are NOT the same thing); the trial staff recruiting patients should NOT have the randomization list. Randomization should be prepared by the trial statistician, and he/she would not participate in the recruiting.

Response:

We have taken your suggestions and the paragraph was modified in the material and methods section:

“The database of 120 patients diagnosed with SSc was accessed by the Rheumatology department, who were contacted by telephone to invite them to participate in this study. 30 patients were accepted and were initially evaluated by Rheumatology department. Then, the patients who met the selection criteria were evaluated by Cardiology, Reproductive Physiology, Internal Medicine, Oncology, Rheumatology and Plastic Surgery to detect contraindications. Of them 20 patients were selected (Fig 1). The epidemiologist and the principal author used a random number table to randomize patients. These patients were allocated into control or experimental groups by the remaining researchers. The allocation was 1:1 ratio. into different groups.”

(b) Sample size/power: It appeared strange to find no paragraph of the desired sample size/power for the study (given that this is analysis of data generated from a randomized trial). Formal power calculation should be presented, should focus on the primary response variable, at some desired effect size, and say at 5% level of significance.

Response:

This is a pilot study and for this reason our “n” is small. Additionally our main goal was to prove the safety of the ADSVF injection into the hand of patients with SSc. Fortunately this was safe and we could continue toward the secondary goals. We must mention that this protocol was designed in 2015.

(c) Statistical Analysis: Mention clearly, why nonparametric assessments (Wilcoxon range test, and Mann-Whitney U tests) were chosen wrt. analysis, bypassing parametric modeling, initially. Furthermore, in the longitudinal evaluation, repeated measures ANOVA was used, which is strictly based on Gaussian assumptions. How were the assumptions checked? If those fail, please resort to nonparametric analysis, such as the Friedman's test, etc. 

Response:

After the logarithmic transformation, the normality assumptions were made with the normality tests: Kolmogorov-Smirnov, Shapiro-Wilk and Levene's Statistic. Verifying the assumptions of normality, for which we proceeded to perform ANOVA of repeated measures. It is added in the statistical analysis: "Logarithmic transformation was performed before such analysis; subsequently, tests of normality (Kolmogorov-Smirnof) and homoscedasticity (Levene's test) were applied to determine the nature of the data."

I have an additional question. Given that the study is longitudinal, with covariates measured either at baseline, or at various time-points, why was a formal longitudinal analysis not conducted, via a linear mixed model, or GEE. 

Response:

We appreciate your comment. Mixed linear models were not made because the sample size is not sufficient to carry out this analysis since it is a pilot study.

3. Results & Conclusions:

(a) The authors should check that any statement of significance should be followed by a p-value in the entire Results section.

Response: 

Dear reviewer, thank you for your observation, we have added in the results section the “p” values that were missing. These are marked in red. 

(b) The Discussion section should clearly state that the findings of this study is only from a RCT of Mexican subjects. Hence, future studies (using subjects recruited at other locations/country) are needed to validate the current findings.

Response: 

It has been an important observation and therefore we have added the following paragraph in discussion:

“We must point out that this study was carried out exclusively in Mexican mestizo patients and despite this, the results are similar to those previously reported. However, it is important to mention that in future studies ethnicity should be a variable to consider.”

Reviewer #3

Major critiques:

1. This is the first noted report of use of lipoaspirate with SVF supplementation to treat systemic sclerosis related hand debility, but the logical jump to utilizing this therapeutic strategy is unclear. Though reported use of SVF demonstrated acute effects, results were not durable, thus alternative strategies were needed. However, two discussed reports of use of fat injections for Ssc seemed to have sustained results, thus it is unclear which aspect of these protocols require improvement. As previous reports suggest long term, durable results were achieved when treated Ssc related hands with lipoaspirate (condensed or decanted), what is the rationale for modifying lipoaspirate in this study? 

Response:

The use of fat grafts (lipoaspirate) on one hand and the use of ADSVF for the treatment of hand deformities caused by SSc on the other were reported simultaneously in 2015. The results reported in both procedures are encouraging. The biological mechanism by which SSc patients that were treated with ADSVF improve is unknown, but improvement is invoked through the regenerative effects of the injected cells. Plastic surgery uses fat grafts with good results for volumetric effects. Therefore, this concept was used in these patients with SSc since they require greater volume on their bony prominences, especially at the dorsal level. Thus we decided to unite both concepts which are already used in plastic surgery under the name of cell-assisted lipotransfer (CAL). Therefore the reasons why we used this combination were to increase the volume of fat tissue, increase the percentage of integration of fat grafts with the administration of ADSVF and to use the regenerative properties of ADSVF.

We have added in the discussion section the following paragraph:

“The mixture of fat micrografts plus ADSVF has previously been used in plastic surgery with the aim of increasing the integration of fat grafts. This procedure has been called cell-assisted lipotransfer. For this reason we justify its use in these patients [38].” 

2. The primary hypothesis for admixing SVF with lipoaspirate in this clinical study is that adipose derived from Ssc patients incurs pathogenic related stem cell deficiencies in native tissue, thus supplementation is necessary. However, the ratio of ASC per gram of lipo in donors is not reported in this study and it is not clear if the naive lipo is deficient in stem cells. Further, the authors determined the total number of nucleated cells added to lipoaspirate prior to injection, yet, it seems no significant correlation existed between supplemented values and measured outcomes. Therefore, it is unclear what value SVF supplementation adds to use of lipoaspirate alone. 

Response: 

Your comment is wise, and we appreciate it. We base ourselves on the study reported by Griffin et al, who pointed out that there was no statistical difference in the number of cells that adhered, alteration in the phenotype or surface antigen expression between adipose derived stem cells of patients with SSc and adipose derived stem cells of healthy patients. Griffin M, Ryan CM, Pathan O, Abraham D, Denton CP, Butler PEM. Characteristics of human adipose derived stem cells in scleroderma in comparison to sex and age matched normal controls: implications for regenerative medicine. Stem Cell Res Ther 2017;8:23-34. doi: 10.1186/s13287-016-0444-7 

Based on the previous report, it was not our objective to evaluate the amount of stem cells that were in the liposuctioned tissue. Our objective in adding ADSVF to the fat grafts was to potentiate the integration of adipocytes as well as to maintain the regenerative properties of ADSVF.

Therefore we have added the following paragraph in the introduction section:

“However, there was no statistical difference in the number of cells that adhered, alteration in the phenotype or surface antigen expression between adipose derived stem cells of patients with SSc and adipose derived stem cells of healthy patients [15]”

3. Development of a surgical or therapeutic approach to improve patient quality of life and reduce hand disability is a critical unmet need, however a future opportunity for a more robust study would be ASC supplementation in a randomly selected hand compared to fat alone in the contralateral, such that patients served as their own internal control to measure the benefit of SVF supplementation. The process of isolation SVF is time consuming and expensive, requiring patients to be exposed to increased duration of anesthesia and surgical risk. Thus, it is extremely important to adequately assess the necessity of the SVF isolation procedure and supplementation. However, as reported herein, this study does not make clear or not, the necessity of SVF supplementation to achieve durable results.

Response:

Dear reviewer, the use of ASC is approved by the Instituto Nacional de Ciencias Medicas y Nutricion Salvador Zubiran’s ethics committee only for research purposes with solid bibliographic records, which did not exist and do not exist for the treatment of these lesions in patients with SSc. ADSVF is a minimally manipulated cell mixture that has been approved for research studies in several countries and therefore in ours. Fatty micrografts have proven their usefulness for several years in plastic surgery. The combination of both concepts (cell-assisted lipotransfer) is a recently used plastic surgery practice. That is why it was used in this way in our patients. Obtaining fat micrografts is performed exclusively with local anesthesia, which significantly reduces the risks and costs for the patient. The processing of the ADSVF was 60 minutes and the entire procedure for obtaining fat grafts and their application to the hands was 120 minutes.

Minor criticisms:

1. The term Wilcoxon "Range" test is used multiple times. Do the authors mean Wilcoxon "Rank" test meant?

Response: 

Dear reviewer, we apologize, definitely the correct term is “rank”, we have changed the word “range” to “rank”, thank you.

2. The authors measured clinical outcomes in the treated hand and had an untreated hand in the same patient which could have served as an internal baseline for the effects of continued medical treatment. In essence, what effects were seen in untreated contralateral hands? 

Response: 

Dear reviewer, the objective of the study was to evaluate the evolution of the right hand treated in the experimental group vs. the right hand of a patient in the control group. The reason why the contralateral hand of the experimental group was not systematically evaluated was due to the uncertainty of whether the application of ADSVF could have a systemic biological or psychological effect in the experimental patient.

Despite this, the most important findings were found when comparing the treated hand vs. the untreated hand in the experimental group: the treated hand presented greater volume than the contralateral hand; higher temperature and faster capillary refill vs contralateral. These findings are described in the text under results:

“Clinically, the treated hands of the experimental group patients exhibited larger volumes, warmer temperatures, and faster capillary filling than the contralateral hands of the same patients (Fig 5 and S1 Video).”

3. More information needs to be provided as to how lipoaspirate was condensed or prepared in the OR prior to reinjection? When lipo washes were performed, which devices or methods were used? How was lipo condensed in the OR and exactly how was the SVF mixed with fat prior to injection? 

Response:

Dear reviewer, the extraction of fat grafts through liposuction was performed under local anesthesia with a Klein’s formula in a minor surgical procedures room equipped with necessary items for emergency situations. ADSVF processing was performed in a cell biology laboratory. The mixture of both compounds was carried out in the minor surgical procedures room where the fat was extracted and it was injected right there into the hands of the patients under local anesthesia.

We have answered question number 4 of reviewer 1 in which we mentioned how and where the mixture of the micrografts with the ADSVF was done:

“After the fat was processed, 2 ml of the ADSVF was transferred back to the operating room and was mixed with the 40 cc of fat in a 50 ml syringe. This mixture was homogenized with a slight and constant manual agitation of the syringe, and later it was transferred to a 3 ml syringe for the autologous transplantation procedure.”

We appreciate your comment and we have changed the word operating room in our text to “minor surgical procedures room”

Reviewer #4

Title

1) As both groups continue their medication, I do not believe that ADSVF vs. medical treatment is an appropriate wording. I suggest removing “versus medical treatment” from the title.

Response:

Dear reviewer, we accepted your suggestion and we have changed the title.

“Adipose derived stromal vascular fraction and fat graft for treating the hands of patients with systemic sclerosis. A randomized clinical trial.” 

Abstract

2) Headlines (Background, Methods, Results and Conclusion) would make the abstract more presentable.

Response: 

Dear reviewer, thank you for your comment, we have restructured the abstract section, please review it in the manuscript.

Methods

3) Please report the reasons for declination of participation. As 100 patients declined it would be relevant to explore a potential selection bias. 

Response: 

Of the 120 patients, only 30 fulfilled the protocol selection criteria investigated by telephone. These 30 patients were subsequently evaluated by different medical specialties for thorough screening. In 10 of them, a family history of allergies, infections and history of cancer with a family tendency were detected, which finally ruled them out from being included in the study. Only one patient in the control group declined due to difficulty in transporting from home to the hospital. 

4) Please clarify how the outcomes were collected. Which outcomes were self-reported? How often did the patients meet for clinical controls? 

Response: 

According to our protocol, all surgically operated patients were reviewed every week during the first postoperative month. Subsequently, they were evaluated on days 56, 84, 112, 140 and 168. Table 1 is a summary to indicate the evaluation days. Rheumatology data was retrieved by the rheumatology group and internal physicians were trained to collect other data. None outcomes were self-reported. The data was collected in an excel format which is in the supplementary material according to the recommendations of the epidemiologist. (See supplementary material number S4 Table). Table 1 as well as table S4 shows how often (0, 28, 56, 84, 112, 140 and 168 days) the control group patients were evaluated. 

5) In line 117 it is stated that a broad group of physicians were involved to detect contraindications. What were the contraindications of the 10 excluded patients? 

Response:

Contraindications in the excluded patients were: allergies, infections, and a history of cancer with a familial tendency.

6) The period of the study is repeated in in line 146 and 202 where September and October are both mentioned as the starting month.

Response: 

We apologize for this mistake, the right month is September. In line 202 the word “October” was changed for “September”.

7) In general, I would suggest using months as time points as 168 days seems more arbitrary.

Response:

We agree with your suggestion, thank you very much, however, our evaluations were already carried out on the indicated days and we consider it correct to describe them as they were done.

Statistics

8) In line 206 it is reported that continuous variables are reported as median with 95%CI. It should either be median with IQR or mean with 95%CI depending on the normal distribution.

Response: 

Dear reviewer, we agree and we added to the statistical analysis section as follows: “Continuous variables are expressed as median with 95% CI with non normal distribution or mean with 95% CI with the normal distribution.” 

9) Age and FTP are reported as means with 95%CI suggesting a normal distribution, however the mentioned statistical tests are usually applied for non-normal continuous outcomes. 

Response:

We apologize for this mistake, and we have changed the follow paragraph in the statistical analysis section:

“The differences between before and after the intervention were analyzed with the Wilcoxon rank test, and the differences between the control and experimental groups at 0 days and 168 days were analyzed with the Mann–Whitney U test with non normal distribution and the student's t test of independent samples with normal distribution.”

And on the footnote on Table 2 has been changed as follow:

 “*Statistical significance analyzed with a student's t test of independent samples.”

10) Where there any patients with multiple ulcers and if yes how was this handled statistically? 

Response: 

Ulcers were counted by units in each group, regardless of how many existed in each patient. At the end, only the number of ulcers was evaluated, not patients. 

Only one patient from each group presented two ulcers each. This did not imply a difference between groups.

Results

11) The results section could be more structured as there are many different outcomes. Either start with the significant results or choose another more appropriate order.

Response:

We appreciate your comment, which is very valuable, however changing the results as you suggest would imply major changes to all the tables and supplementary material. We kindly ask you to allow us to present it in the way we have done so.

12) When reporting p-values it should be clearer whether the analysis is a within group versus between group analysis.

Response:

Dear reviewer, thank you for your suggestion, we have corrected the footnote in Table 4 as follow: 

“P1= Analyze the differences within each group between baseline and final results. Wilcoxon signed-rank test was used.

*p<0.05, U de Mann-Whitney with statistical significance. analyzes the differences between groups.

P2= These data were log-transformed before statistical analysis was ANOVA for repeated measures to determine the time x group interaction. Analyze the differences between the groups.“

13) In table 2 I would suggest not to perform statistical comparisons in groups without any values (e.g. severe cardiac involment) as a comparison of zero-values is not meaningful. 

Response:

Indeed, the statistical comparison between groups of the severity of internal organs of SSc does not seem to have any relevance, however, this comparison allows us to demonstrate that both groups are similar in the severity of the disease. We kindly ask you to allow us to present this data that does not distract or affect the final result.

14) All p-values should be reported as less than instead of equals and follow the system of p<0.05, p<0.01, p<0.001 and p<0.0001. Non-significant p-values should be exact. 

Response: 

Dear reviewer, we apologize for our mistake and we have changed the symbol = by <. 

15) In the legend it states that the reported p-values in p1 are both within and between group analyses but only two p-values are reported (if both within and between group analyses are reported I would expect three p-values?) 

Dear reviewer, we designate the comparison of the evolution within the experimental group = p1, in this same way the comparison of results in the control group was made and we also call it p1. P2 was between groups. You are correct, there was a mistake in the footnote of Table 4 which we have corrected:

“P1= Analyze the differences within each group between baseline and final results. Wilcoxon signed-rank test was used.

*p<0.05, U de Mann-Whitney with statistical significance. analyzes the differences between groups.

P2= These data were log-transformed before statistical analysis was ANOVA for repeated measures to determine the time x group interaction. Analyze the differences between the groups.“

16) There is a mismatch between the values provided in table 4 and figure 4. E.g. the pain score in the experimental group is reported as 5.0 and 0 in table 4 but is visually assessed to be 4.5 and 1.5 in figure 4.

Response:

Dear reviewer, both pain values in Table 4 and Figure 4 are indeed different, due to the fact that the statistical method used to assess pain was different, as mentioned in the manuscript. Indeed you are right and we have changed the Figure 4 with the values written in Table 4, which were evaluated with the Wilcoxon signed-rank test and ANOVA.

We have made changes in the following paragraph:

“For a better visual perception of pain evolution we have made a graphic representation , this was analyzed with a descriptive method and standard deviation (Fig 4).”

We have also changed the legend in Figure 4:

“Figure 4. Graphic evolution of pain in the control and experimental group, evaluated with the Wilcoxon signed-rank test and ANOVA.”

Discussion

17) I would suggest shortening the discussion and be more selective and concise when commenting your own results with the literature as reference.

Response:

Dear reviewer, we understand your point of view on how to structure the discussion. However, in the discussion section, we have wanted to explain the uncertainty that exists when applying fat grafts, decanted fat grafts or the application of ADSVF and the results obtained by the different authors. Thus, we have subsequently mentioned our results so that the reader can evaluate which method may be the best. We appreciate your intention to improve the paper but we ask you to allow us to present it as it is structured.

18) I suggest beginning the discussion with the key results from this study with a subsequent comparison with the literature.

Response:

Dear reviewer, we understand your point of view on how to structure the discussion. However, in the discussion section, we have wanted to explain the uncertainty that exists when applying fat grafts, decanted fat grafts or the application of ADSVF and the results obtained by the different authors. Thus, we have subsequently mentioned our results so that the reader can evaluate which method may be the best. We appreciate your intention to improve the paper but we ask you to allow us to present it as it is structured.

19) There should be a limitation section. Despite the randomized design the study is still of low quality due to a small sample size, no blinding of patients, surgeons or outcome assessors, no sham-procedure, loss to follow-up of 10% in one group and no power-calculation. These limitations should be stated.

Response:

We appreciate your trascendental suggestion and have added the following to the end of the discussion:

“This study has the disadvantages of not having been a blind study for patients and researchers, as well as a limited number of patients studied. The loss of a patient even though he belonged to the control group is a significant loss considering the size of the sample. Future studies should try to avoid these limitations.”

20) In line 372-374: Why did the medical treatment (which was also continued in the experimental group), the warmer environment not affect the experimental group? 

Response:

Dear reviewer, without a doubt the medical treatment and the changes in environmental temperature could have had a positive effect on the patients. But this effect happened in patients from both the experimental group and the control group. The above is our assumption. However, the variable between both groups was the application of ADSVF plus fat grafts that we considered were the cause of the improvement of the variables with statistical significance in the experimental group.

---

## [Decision Letter · Decision Letter 1]

16 Jun 2023

PONE-D-23-00573R1Adipose derived stromal vascular fraction and fat graft for treating the hands of patients with systemic sclerosis. A randomized clinical trial.PLOS ONE

Dear Dr. Iglesias, Thank you for submitting your manuscript to PLOS ONE. After careful consideration, we feel that it has merit but does not fully meet PLOS ONE’s publication criteria as it currently stands. Therefore, we invite you to submit a revised version of the manuscript that addresses the points raised during the review process.

We look forward to receiving your revised manuscript.

Kind regards,

Ruochen Dong, M.D./Ph.D.

Guest Editor

PLOS ONE

Journal Requirements:

**Additional Editor Comments:** Your revised manuscript has been evaluated by three reviewers, and their comments are available below. In general, the majority of reviewers thought your responses and revision addressed their concerns. However, reviewer #4 still raised concerns. After carefully reviewing the revised manuscript and the reviews' comments, I would suggest a minor revision for your manuscript.  

Please note that the following changes are **required for acceptance**:

Please have figure legends of each figure with proper descriptions. All figure legends should be listed in a "Figure Legends" section.Please round the p-value in the **main text **to the nearest alpha-threshold (0.05, 0.01, 0.001, 0.0001, etc.). The p-value in the tables could still use exact numbers and be highlighted with asterisks to show the significance.

Please note that the following changes are **suggested but not required for acceptance**:

Please use box plots or violin plots to interpret each data point in Figure 4. The current interpretation is misleading as the reader may think it represents mean ± SD/SEM instead of median ± 95%CI.Shorten the discussion.

Reviewers' comments:

Reviewer's Responses to Questions

**Comments to the Author**

1. If the authors have adequately addressed your comments raised in a previous round of review and you feel that this manuscript is now acceptable for publication, you may indicate that here to bypass the “Comments to the Author” section, enter your conflict of interest statement in the “Confidential to Editor” section, and submit your "Accept" recommendation.

Reviewer #1: All comments have been addressed

Reviewer #2: All comments have been addressed

Reviewer #4: (No Response)

2. Is the manuscript technically sound, and do the data support the conclusions?

Reviewer #1: Yes

Reviewer #2: (No Response)

Reviewer #4: No

3. Has the statistical analysis been performed appropriately and rigorously? 

Reviewer #1: Yes

Reviewer #2: (No Response)

Reviewer #4: No

4. Have the authors made all data underlying the findings in their manuscript fully available?

Reviewer #1: Yes

Reviewer #2: (No Response)

Reviewer #4: Yes

5. Is the manuscript presented in an intelligible fashion and written in standard English?

Reviewer #1: Yes

Reviewer #2: (No Response)

Reviewer #4: No

6. Review Comments to the Author

Reviewer #1: This study is very interesting and the authors have responded to all requests.

I have no additional request

Reviewer #2: (No Response)

Reviewer #4: The reviewer thanks the authors for their revisions of the manuscript and for the provisions of multiple clarifications.

1)

I firmly believe that the discussion section can be structured more appropriately. I have a concern that a discussion section of more than six A4 pages is far too excessive to maintain the focus of the reader. I would still suggest shortening of this section.

2)

As an extension to my previous point regarding p-values: Except for changing = to < the p-values are still reported as exact (e.g. p<0.003). P-values are conventionally reported as clearly defined alpha-thresholds (0.05, 0.01, 0.001, 0.0001 etc.). Therefore, all reported p-values should be rounded to the nearest alpha-threshold.

3)

In figure 4 the standard deviations are clearly asymmetrical. This raises my concerns regarding the statistical analysis as standard deviations are normally symmetrical around the mean. Either the standard deviations have not been calculated correctly or it is instead an interquartile range that has been visualized in the plot. If this is the case, the assumptions for a repeated measures ANOVA have been violated.

4)

The authors report that in case of multiple ulcers, the sample unit was the number of ulcers and not patients with ulcers. This ignores the effect of clustering (as multiple ulcers are clustered within the same patient). This is pseudoreplication which increases the risk of type I-errors due to falsely low variances.

Either 1) an appropriate statistical method should be used to handle this scenario, 2) the sample unit should be the number of patients with one or more ulcers or 3) mention this limitation in the limitation section with a phrase that the results should be interpreted with caution.

7. PLOS authors have the option to publish the peer review history of their article (what does this mean?). If published, this will include your full peer review and any attached files.

Reviewer #1: No

Reviewer #2: No

Reviewer #4: No

---

## [Author Response · Author response to Decision Letter 1]

15 Jul 2023

LETTER 2 OF RESPONSES TO REVIEWERS

Dear editor:

We are very happy to hear about the publisher's decision. We send the responses to the reviewers and hope to meet all your requirements.

Response to editor

Comment 1. Dear editor, we have written the manuscript according to the journal's instructions. However, as you have suggested, we have added to the end of the references a section called Figure Legends.

Comment 2. The p-value in the manuscript has been changed as you have suggested.

Changes suggested. 

Comment 1. The figure 4 has been modified using box plots, and the statistical analysis was performed with Wilcoxon signed rank test.

Comment 2. The discussion has been shortened, and the references have been restructured.

Response to reviewers

Reviewer 4.

Comment 1. The discussion has been modified and shortened and the references have been restructured.

Comment 2. The p value in the manuscript has been modified according to your suggestions.

Comment 3. The figure 4 has been modified using box plots and the statistical analysis was performed with Wilcoxon signed rank test.

Comment 4. Thank you for your comment. We agree with your point of view. Due to an error in the initial capture of data on the number of ulcers per patient, we were unable to know how many initial and final ulcers there were in a patient. Therefore, in the discussion section it has been pointed out as a limitation as follows:

“Although the number of digital ulcers was statistically lower in the experimental group compared to the control group at the end of the study, these results must be interpreted with caution, since unfortunately we did not assess the number of digital ulcers existing at the beginning and at the end of the study in each patient.”

---

## [Editor Report · Decision Letter 2]

24 Jul 2023

Adipose derived stromal vascular fraction and fat graft for treating the hands of patients with systemic sclerosis. A randomized clinical trial.

PONE-D-23-00573R2

Dear Dr. Iglesias,

We’re pleased to inform you that your manuscript has been judged scientifically suitable for publication and will be formally accepted for publication once it meets all outstanding technical requirements.

Kind regards,

Ruochen Dong, M.D./Ph.D.

Guest Editor

PLOS ONE

---

## [Editor Report · Acceptance letter]

31 Jul 2023

PONE-D-23-00573R2 

Adipose derived stromal vascular fraction and fat graft for treating the hands of patients with systemic sclerosis. A randomized clinical trial. 

Dear Dr. Iglesias:

I'm pleased to inform you that your manuscript has been deemed suitable for publication in PLOS ONE. Congratulations! Your manuscript is now with our production department. 

Kind regards, 

on behalf of

Dr. Ruochen Dong 

Guest Editor

PLOS ONE